# Transcriptome Analyses in a Selected Gene Set Indicate Alternative Oxidase (AOX) and Early Enhanced Fermentation as Critical for Salinity Tolerance in Rice

**DOI:** 10.3390/plants11162145

**Published:** 2022-08-18

**Authors:** Shahid Aziz, Thais Andrade Germano, Karine Leitão Lima Thiers, Mathias Coelho Batista, Rafael de Souza Miranda, Birgit Arnholdt-Schmitt, Jose Helio Costa

**Affiliations:** 1Functional Genomics and Bioinformatics, Department of Biochemistry and Molecular Biology, Federal University of Ceara, Fortaleza 60451-970, Ceara, Brazil; 2Non-Institutional Competence Focus (NICFocus) ‘Functional Cell Reprogramming and Organism Plasticity’ (FunCROP) (Coordinated from Foros de Vale de Figueira), 7050-704 Alentejo, Portugal; 3Postgraduate Program in Agricultural Sciences, Campus Professora Cinobelina Elvas, Federal University of Piauí, Bom Jesus 64900-000, Piaui, Brazil

**Keywords:** cell reprogramming, ROS formation control, Pokkali, IR29, crop development

## Abstract

Plants subjected to stress need to respond rapidly and efficiently to acclimatize and survive. In this paper, we investigated a selected gene set potentially involved in early cell reprogramming in two rice genotypes with contrasting salinity tolerance (Pokkali tolerant and IR29 susceptible) in order to advance knowledge of early molecular mechanisms of rice in dealing with salt stress. Selected genes were evaluated in available transcriptomic data over a short period of 24 h and involved enzymes that avoid ROS formation (AOX, UCP and PTOX), impact ATP production (PFK, ADH and COX) or relate to the antioxidant system. Higher transcript accumulation of AOX (ROS balancing), PFK and ADH (alcohol fermentation) was detected in the tolerant genotype, while the sensitive genotype revealed higher UCP and PTOX transcript levels, indicating a predominant role for early transcription of AOX and fermentation in conferring salt stress tolerance to rice. Antioxidant gene analyses supported higher oxidative stress in IR29, with transcript increases of cytosolic CAT and SOD from all cell compartments (cytoplasm, peroxisome, chloroplast and mitochondria). In contrast, Pokkali increased mRNA levels from the AsA-GSH cycle as cytosolic/mitochondrial DHAR was involved in ascorbate recovery. In addition, these responses occurred from 2 h in IR29 and 10 h in Pokkali, indicating early but ineffective antioxidant activity in the susceptible genotype. Overall, our data suggest that AOX and ADH can play a critical role during early cell reprogramming for improving salt stress tolerance by efficiently controlling ROS formation in mitochondria. We discuss our results in relation to gene engineering and editing approaches to develop salinity-tolerant crops.

## 1. Introduction

Salinity is the major impediment shattering the productivity of cultivated land areas. Excessive salt accumulation causes severe ionic toxicity, increases soil compactness, reduces plants’ ability to acquire water and obstructs efficient transportation of nutrients, thus interfering with crop production and yields worldwide [1]. Rice (*Oryza sativa*), a model cereal crop, is a premier staple food, providing a large proportion of the human population’s food and income for billions across the globe. Rice is categorized as a typical glycophyte and is vulnerable to climate changes, thus harming food security [2,3]. However, abundant natural variability and various cultivated rice genotypes demonstrate contrasting responses to salt stress. Pokkali can withstand salinity, which is used as a positive control in screening salt-tolerant rice cultivars, while IR29 is considerably salt-sensitive and used as negative control [4]. Understanding the genes and mechanisms that regulate environmental stress in crops is critical for boosting agricultural yield and quality, safeguarding food security and even protecting important crops from extinction. Comparative analysis of stress-responding genes and their interconnected networks in rice genotypes with contrasting responses to salinity stress may lead to better comprehension of salinity-tolerating mechanisms and the identification of relevant genes for molecular breeding.

Tolerance of stresses is a complex phenomenon involving several particular gene loci with distinct regulation, molecular aspects and an array of interconnected mechanisms that maintain plant homeostasis upon exposure to hostile conditions [5]. In general, stress tolerance is linked to the maintenance of cellular redox homeostasis, regulating the levels of reactive oxygen species (ROS) required to initiate biological processes and function as signaling molecules to trigger plant defense responses [6]. Plants have different systems that act to regulate ROS formation by using alternative oxidase (AOX), uncoupling protein (UCP) and plastid terminal oxidase (PTOX) or ROS scavenging by enzymatic and non-enzymatic antioxidants.

The inner-facial mitochondrial membrane of plant cells harbors energy-dissipating alternative respiratory systems mediated by AOX. The AOX gene family in angiosperms is nucleus-encoded, composed of one to six members in two subfamilies (AOX1 and AOX2) linked to stress and housekeeping functions [7,8,9,10]. However, in monocots, the AOX2 subfamily is restricted only to some species of the Alismatales order [10], while most studies show that monocots have four or five AOX1 genes [9]. In rice, four AOX1 genes (*AOX1a*, *1c*, *1d* and *1e*) have been found [9,11]; *AOX1a* and/or *AOX1b* (renamed to *AOX1d*) in [9] were induced by different stress conditions such as chilling, drought and high salt, while *AOX1c* was stably detected, and *AOX1e* was barely expressed in germinating seeds [9,12,13,14,15,16,17,18]. In stress conditions, AOX relaxes the highly coupled and tensed electron transport process by driving electrons from quinol to oxygen, thereby alleviating tensed conditions and reducing ROS production [19]. These characteristics may allow plants to flexibly deal with the challenge of changing scenarios and induce plasticity, facilitating plant persistence.

In addition to AOX, the plant mitochondrial inner membrane possesses UCPs. The UCPs belong to the superfamily of mitochondrial carrier proteins dissipating the proton electrochemical gradient generated by the respiratory chain complexes [20]. In plants, these proteins are involved in mitochondrial energy flow regulation. They have been suggested to play a critical role in mitigating ROS production by the mitochondrial electron transport chain [21]. Moreover, another terminal oxidase is PTOX, which is located in chloroplasts. PTOX is a key factor for maintaining the plastoquinone (PQ) pool redox balance and functions as a “safety valve” to protect photosynthesis [22]. It is a stress-responsive protein and could protect plants from various harmful stresses [23].

With changing environments, the AOX, PTOX and UCP genes show differential expression patterns and are induced by multiple signaling pathways [24,25,26]. Together, AOX, UCP and PTOX are considered primary defense lines mitigating ROS production, an excess of which causes progressive oxidative damage and ultimately cell death. Thus, these protein systems allow for flexibly dealing with the challenge of several stressors, restoring respiratory activities and correcting metabolism.

Cellular damage manifests when the delicate balance between ROS production and elimination is disturbed upon exposure to severe stress. To minimize the damaging effect of ROS, plants have developed an efficient antioxidant system with two components: enzymatic and non-enzymatic antioxidants. In plants, the non-enzymatic antioxidant ROS-scavenging pathway involves ascorbate and glutathione metabolites mediated by the ascorbate–glutathione cycle (AsA-GSH) in chloroplasts, cytosol, mitochondria and peroxisomes [27,28]. Enzymatic components of the antioxidant defense system comprise several antioxidant enzymes such as superoxide dismutase (SOD), catalase (CAT) and glutathione peroxidase (GPX), which catalyze ROS degradation; and enzymes of the ascorbate–glutathione (AsA-GSH) cycle, such as ascorbate peroxidase (APX), monodehydroascorbate reductase (MDAR), dehydroascorbate reductase (DHAR) and glutathione reductase (GR), that regenerate soluble antioxidants [29,30].

To mitigate and recover from the damaging effects of adverse environmental conditions, understanding and developing mechanisms such as effective reprogramming of a damaged cell is among the primordial needs. Induced cell reprogramming permanently facilitates plants’ immediate persistence regarding environmental factors’ variability throughout the lifetime. Consequently, it promotes individual cell growth and organism survival. Thus, early reprogramming in response to multiple individual and combined stressors could be a unique positive response across plant species and even across diverse taxonomic classes [31,32,33]. AOX has demonstrated a significant role in plant homeostasis, reprogramming and plant growth adaptation in response to diverse abiotic and biotic stresses [34,35,36,37,38]. AOX has positive physiological roles in certain developmental processes and adaptation to environmental stresses. In doing so, AOX improved the ability of cells to rapidly recover their energy status [19]. Short- and long-term fine-tuning of AOX at the transcriptional level was essential for positive performance effects [7,39]. Finally, adaptive plant robustness in the field was shown to connect to the capacity for efficient cell reprogramming, which could be measured already at the level of seeds [34,38]. Prediction of plant holobiont robustness could be linked in a technically simple way to AOX by inhibiting its activity. These tests promoted its use in seed screening for diverse species and also for low-cost *on-farm* seed selection, and they are awaiting broader validation [34,38,40,41,42].

Recently, our group demonstrated that transcript accumulation of genes linked to early cell reprogramming under stress related primarily to ROS/RNS balancing and energy status connected to cell restructuration and cell cycle regulation [33,43,44,45,46]. Our approach is in conformity with the view that “optimization of adaptive potential requires reconfiguration of developmental attributes to allow growth adjustment and stress avoidance” [47]. Thus, in the current study, we explored a selected gene set in public transcriptomic data of two rice cultivars with contrasting responses to salt stress (Pokkali tolerant and IR29 susceptible) during 24 h following salt stress treatment to advance knowledge on the relevance of early cell reprogramming to general plant plasticity and robustness. Pokkali is most famous for salt tolerance but is appropriate for our approach because this traditional cultivar has a broad spectrum of resilience [48,49]. Here, we focused on gene expression involved in ROS formation (AOX, UCP and PTOX), impacting ATP production (PFK, ADH and COX) and associated with the antioxidant systems (APX, MDHAR, DHAR, GR, CAT, SOD and GPX) in different cell compartments in order to gain insight into early cell reprogramming of salinity tolerance in rice. The results are discussed in relation to a connected view of redox homeostasis and energy supply as critical traits for salinity tolerance.

## 2. Material and Methods

### 2.1. Gene Expression Analyses of RNA-Seq Data

This study used publicly available RNA-seq data of two rice genotypes with contrasting responses to salt stress (Pokkali tolerant and IR29 susceptible) [50]. Both genotypes were grown in growth chambers to the three-leaf stage. The salt stress treatment was applied by watering 2-week-old seedlings with 300 mM NaCl solution or by normal watering in control plants. Shoots (stem and leaves) were harvested at 1, 2, 5, 10 and 24 h post-treatment to obtain transcriptomic data [50]. The transcriptomic data are available in SRA database from Genbank (NCBI) under the following Bioproject numbers: PRJEB4671 (Pokkali) and PRJEB4672 (IR29).

The expression analysis of target genes in transcriptomic data, with three replicates for each sample, was performed in three steps: (1) mapping of reads by the Magic-Blast software [51]; (2) quantification of mapped reads using the HTseq program [52]; and (3) normalization of read amount in all samples. Thus, in the mapping of the reads, the target cDNAs were aligned against RNA-seq data. After quantification of the mapped reads, the normalization of reads among different samples was carried out using the RPKM (reads per kilobase of transcript per million mapped reads) method [53] according to the following equation: RPKM = (number of mapped reads × 10^9^)/(number of sequences in each database X number of nucleotides of each gene).

The target genes were associated with glycolysis, fermentation, aerobic respiration, anti-ROS formation and antioxidants. Thus, these genes included seven gene members encoding cytosolic PFK (phosphofructokinase) [54] to represent total PFK in glycolysis. Four ADH (alcohol dehydrogenase) genes [55] denoted total ADH in alcohol fermentation, while mitochondrial aerobic respiration was represented by total COX (cytochrome c oxidase), with eleven gene members [56]. In addition, total AOX (alternative oxidase) with four genes [9,11] and UCP (UCP 1 and 2) [57] were the systems involved in mitochondrial ROS formation control, while a single PTOX gene, as per Tamiru et al. [58], represented the system regulating ROS formation in the chloroplast (Appendix A). Regarding antioxidant enzymes, multiple gene families of APX, MDHAR, DHAR, GR, SOD, GPX and CAT that encode proteins with different subcellular destinations, such as cytosol, peroxisomes, mitochondria and chloroplasts (Appendix A), were evaluated. In general, all antioxidant genes presented at least one member associated with each compartment, except GPX (without members related to cytosol) and CAT (without members related to chloroplasts and mitochondria). To infer the glycolysis pathway, we analyzed genes encoding phosphofructokinase (PFK) enzyme, as it is well known that its regulation is highly influenced by the energy status of the cell.

### 2.2. Prediction of Subcellular Localization of Antioxidant Proteins

For the prediction of the subcellular localization from corresponding deduced antioxidant proteins (listed in Appendix A), the following tools were used: TargetP-2.0. Available online: http://www.cbs.dtu.dk/services/TargetP (accessed on 20 March 2022). MitoProtII. Available online: https://ihg.gsf.de/ihg/mitoprot.html (accessed on 22 March 2022). DeepLoc-1.0. Available online: http://www.cbs.dtu.dk/services/DeepLoc (accessed on 24 March 2022) and Plant-mSubP. Available online: http://bioinfo.usu.edu/Plant-mSubP (accessed on 25 March 2022) [59,60,61]. In addition, experimental confirmation of subcellular localization was available for proteins of some gene members of APX [62,63] MDHAR [64,65,66] CAT [67], SOD [68].

### 2.3. Statistical Analysis

The statistical analyses of gene expression data were performed using the GraphPad Prism 9.0 software. The results were expressed as means (RPKM values) ± standard deviation (SD) from three biological replicates. The data obtained were subjected to analysis of variance (ANOVA) using the GraphPad Prism 9.0 software, and Bonferroni’s test compared averages at 5% probability.

## 3. Results

### 3.1. Tolerant Genotype Shows Elevated Transcript Levels of PFK (Glycolysis) and ADH (Fermentation)

In Figure 1, total transcript levels of PFK, ADH and COX are shown to indicate the status of glycolysis, fermentation and aerobic respiration in two rice genotypes differing in salt stress tolerance [Pokkali (tolerant) and IR29 (susceptible)].

In general, higher mRNA levels of total PFK and ADH were observed in the tolerant genotype across the time points, with a significant difference in most cases. The only exception was the point salt stress at 24 h, in which similar levels of both transcripts were detected for Pokkali and IR29 genotypes (Figure 1A,B). Regarding the salt stress effect, a significant difference in the controls was found in PFK expression, with a decrease at 5 h in Pokkali and an increase at 24 h in IR29 (Figure 1A,B).

Interestingly, the differential gene expression pattern observed for total PFK and ADH among both rice genotypes differed from the pattern of total COX transcripts (Figure 1C). Overall, similar COX transcript levels were observed in both genotypes across all time points, except in the time point control 10 h. Significantly higher COX mRNA levels were observed in the IR29 genotype. A single significant COX mRNA decrease occurred at 24 h in the Pokkali genotype under salt stress.

### 3.2. Alternative Oxidase Expression Is Preponderant in Tolerant Genotype among Different ROS Formation Control Systems

The total transcript levels of AOX, UCP and PTOX, are shown in Figure 2 to provide insight into energy-dissipating systems in mitochondria (AOX and UCP) and plastids (PTOX) in rice genotypes under salt stress.

Curiously, higher total AOX transcripts (Figure 2A) were observed in the tolerant (Pokkali) genotype compared to the susceptible (IR29) one, with significant data in the majority of cases. This difference can be associated with a higher expression of *AOX1a* (Figure 2D). Concerning the salt stress effect, a total AOX mRNA decrease (significant) was observed at times 1, 5 and 10 h in Pokkali, while some AOX mRNA increase was observed at 24 h in IR29 (not significant) (Figure 2A).

For UCP, higher mRNA levels were detected in the susceptible IR29 genotype than in the tolerant one in both control and salt conditions, although the values were not significant at most time points. No significant salt stress effect was observed (Figure 2B). This difference between genotypes can be connected with higher UCP1 expression in IR29 (Figure 2D).

Concerning PTOX, differences between the two genotypes and two treatments were found mainly at 24 h. At this time point, significantly higher PTOX mRNA levels were observed in IR29 in response to salt stress (Figure 2C).

### 3.3. Antioxidant Gene Expression Indicates Redox Status Compartmentalization in Rice Genotypes under Salinity

The expression analyses of different gene members of APX, MDHAR, DHAR, GR, SOD, GPX and CAT indicated the redox status of different subcellular compartments such as cytosol, peroxisomes, mitochondria and chloroplasts. In general, the main increase responses to salt stress differed between genotypes, with IR29 responding from 2 h while Pokkali responded from 10 h (Figure 3).

For genes encoding proteins to the cytosol, three transcripts (APX, DHAR and SOD) presented significant changes in response to salt stress compared to the control in both genotypes. APX transcripts significantly increased in Pokkali at 10 and 24 h and in IR29 at 2 and 10 h. On the other hand, DHAR increased in Pokkali (1 to 24 h) and decreased in IR29 (1 h), while SOD decreased in Pokkali (1 and 5 h) and increased in IR29 (2 to 24 h). In addition, cytosolic CAT mRNA decreased in Pokkali and increased in IR29 (1 h). Also, cytosolic GPX mRNA increased in Pokkali (non-significant) and remained stable in IR29.

In peroxisomes, both MDHAR and DHAR transcripts significantly decreased in IR29 at 1 h, while DHAR transcripts increased in Pokkali at times 1 to 24 h. In response to salt stress, CAT transcripts increased significantly in Pokkali (24 h) and in IR29 (5 and 24 h). Also, peroxisomal SOD decreases in Pokkali and increases in IR29 were observed (not significant).

In chloroplasts, GPX revealed significant changes in response to salinity in both genotypes. GPX transcripts increased in IR29 at times 2 to 24 h and decreased in Pokkali at 1 to 24 h. In addition, plastid.mito APX transcripts increased significantly in Pokkali (24 h) and decreased in IR29 (2 h). Likewise, SOD increases were observed in both genotypes (not significant). GPX mRNA increases were observed in both genotypes (not significant). In addition, SOD increased in IR29 and decreased in Pokkali (not significant). Plastid.mito DHAR increased in Pokkali (10 and 24 h), while a slight decrease was observed in IR29.

## 4. Discussion

In this research, we support the hypothesis that salt tolerance in rice involves AOX linked to rapidly induced alternative energy production via glycolysis-driven aerobic fermentation. Following the current insight that transcript-level changes during early cell reprogramming can be critically relevant [33,43,44,45] for predicting later performance, we explored salt-induced transcript accumulation changes for the selected gene sets for up to 24 h. Thus, we selected transcriptomic data from two rice genotypes with known contrasting salinity stress tolerances in the field (Pokkali and IR29) treated at the seedling stage with 300 mM NaCl. Recently, our group confirmed that AOX might play a relevant role under mild stress (e.g., at watering for seed germination) and also under severe stress conditions (e.g., induction of somatic embryogenesis) and demonstrated that this was connected to temporarily enhanced fermentation [34,38]. Here, we advanced our knowledge of redox homeostasis (AOX/antioxidant enzymes) and energy supply (glycolysis/fermentation) under severe salt stress in rice. Singh et al. [69] pointed out the importance of mild salt stress as critical for the reproductive stage. However, these authors pointed also to the danger of higher salt concentrations under higher temperatures and low relative humidity when transpiration increases [70].

In fact, the higher transcript levels of AOX, PFK and ADH in the tolerant (Pokkali) compared to susceptible (IR29) genotype (Figure 1 and Figure 2) support a critical involvement of AOX and glycolysis/fermentation in salt stress tolerance. Corroborating these findings, AOX expression variation was also observed in *Vigna unguiculata* cultivars, contrasting in salt/drought stress tolerance [8]. In rice, generally, *AOX1a* and *AOX1d* are the stress-responsive genes (9, 12, 13, 14, 15, 16, 17, 18). However, regarding the present experiment (seedling stage under 300 mM NaCl for 24 h) the higher AOX expression in the tolerant genotype was due to *AOX1a* (Figure 2D). In this regard, very recently, Challabathula et al. [71] also observed this peculiarity of higher *AOX1a* expression in rice cultivars tolerant to drought and salinity. Among other metabolic pathways, glycolysis transcripts increased under salinity in stress-tolerant rice cultivars [72]. More recently, Bharadwaj et al. [38] showed that adaptive reprogramming during early seed germination requires enhanced fermentation, and it involves a critical role of AOX to maintain metabolic homeostasis. Also, Costa et al. [43,44,45] showed that variable ROS/RNS rebalancing and temporarily increased aerobic fermentation appear to generally combine stress-defense mechanisms in humans. These traits are connected to cell restructuration and can discriminate stress factors and distinguish genotypes of cell origins.

Furthermore, Zheng et al. [73] identified AOX pathway involvement with waterlogging tolerance in watermelon, which was associated with increased fermentation instead of aerobic respiration. They compared two contrasting genotypes, YL (tolerant) and Zaojia8424 (sensitive) and observed a strong increase of *AOX* and *ADH* transcripts in the tolerant genotype over 24 h. Also, higher *AOX* and *ADH* mRNA levels were always detected in the tolerant genotype at all analysis times (until 72 h). Considering the relevance of aerobic respiration during early hours after stress perception, these authors observed a strong decrease in *COX* transcripts in both genotypes over up to 24 h of stress. In rice, we detected similar *COX* mRNA levels in both genotypes at the majority of the evaluated time points up to 24 h (Figure 1). Overall, these data support our findings in rice genotypes, indicating a critical role of AOX in stress tolerance followed by efficient respiration and mitigating oxidative impairment in the tolerant genotype (Pokkali).

Because AOX function avoids ROS formation, it is also important to investigate the antioxidant systems among genotypes with contrasting tolerances. According to Lakra et al. [74], the Pokkali genotype has a more efficient antioxidant system than other, salt-susceptible genotypes. In our data, the susceptible IR29 genotype, overall, revealed early antioxidant response from 1 or 2 h compared to Pokkali, which responded from 10 h (Figure 3). This early response could be due to the higher oxidative stress in IR29 compared to Pokkali. In fact, SOD transcripts in IR29 increased in response to salt stress in all cell compartments, while in Pokkali *SOD* expression increased only in the chloroplast (Figure 3 and Figure 4), supporting disseminated O_2_^−^ overproduction in IR29. In this context, it is of interest that Costa et al. [46] observed an earlier and higher transcript increase of ASC-GSH cycle genes in susceptible soybean genotypes compared to the tolerant ones in response to different biotic and abiotic stresses.

In addition, given that APX, GPX and CAT are the main H_2_O_2_ scavenging enzymes in plants [75], our data indicated that rice genotypes under salinity used different enzymatic pathways to control H_2_O_2_ concentration in cell compartments (Figure 4). Apparently, Pokkali preferably activates ASC-GSH cycle genes to scavenge H_2_O_2_ via APX from cytosol and organelles (mitochondria and chloroplasts), while IR29 seems to rather activate the ASC-GSH cycle in cytosol and GPX in organelles (Figure 3 and Figure 4). Also, the ascorbate recycling via ASC-GSH cycle appeared more active in Pokkali because cytosolic/organelles *DHAR* mRNA levels increased only in this genotype (Figure 3). In this regard, stress-tolerant genotypes in *Glycine max* also differed from susceptible genotypes increasing transcripts involved in ascorbate regeneration [46]. Some works showed that *DHAR* overexpression successfully conducted transgenic plants to abiotic stress tolerance, such as in response to aluminum and cold [76]. Among these enzymatic systems, APX appears to be the pivotal antioxidant enzyme to maintain the H_2_O_2_ balance because it has higher H_2_O_2_ affinity, acting the same in very slow protein concentration [77,78], whereas CAT is more associated with H_2_O_2_ detoxification [79]. Thus, while higher CAT could act in peroxisomes, scavenging toxic H_2_O_2_ in both genotypes (Figure 3 and Figure 4), the cytosolic CAT transcript decrease observed only in Pokkali suggests that cytosolic H_2_O_2_ concentration is regulated mainly by the ASC-GSH cycle via APX. However, the higher antioxidant efficiency observed in Pokkali (tolerant) can start much earlier through alternative pathway activity to avoid ROS formation, denoted by the higher AOX mRNA levels observed in this genotype compared to IR29 (susceptible). Supporting our findings, very recently, Challabathula et al. [71] observed that increased *AOX1a* mRNA levels with an efficient antioxidant system were essential for tolerant rice cultivars to maintain lower ROS, higher photosynthesis rates and stress tolerance.

Interestingly, higher *AOX*, *PFK* and *ADH* transcript levels already occur in Pokkali control plants, suggesting that this genotype could also be more resistant to other stress conditions. In this context, some studies show Pokkali as more resistant to lead (Pb) accumulation [48] or as having a similar expression profile of abiotic inducible genes in response to multiple stresses such as NaCl, ABA, polyethylene glycol (PEG) or cold (4 °C) [80]. Thus, Pokkali could have a resourceful genetic background [81] or an intrinsic environmental feature involved in stress tolerance. In this sense, recently, Sampangi-Ramaiah et al. [82] identified a salt-tolerant endophyte (Fusarium sp.,) in Pokkali, which could confer salt tolerance when colonizing the salt-sensitive rice variety IR64. Nevertheless, it was possible to detect this endophyte in the Pokkali genotype in our transcriptomic data, but in a small amount (data not shown). However, we identified Fusarium also in IR29 at a similarly low amount (data not shown). Fungal endophyte diversity in plants depends nonexclusively on genotypes and their effects on the surrounding environment. Natural environmental contexts importantly influence plant endophyte diversity, mainly in the rhizosphere, and thus, depending on agricultural management, also in experimental conditions [83]. Bharadwaj et al. [38] suggested that microbiota can provide a sink for stress-induced higher levels of sucrose and, in this way, might help to alleviate oxidative stress through overloaded mitochondria as a consequence of enhanced glycolysis. In this sense, genotype-compatible endophytes can complement fermentation and alternative respiration through their effect on maintaining host metabolic and energetic homeostasis. Furthermore, it was suggested that endophyte-born AOX genes could complement plant AOX capacities as an added value that evolved under plants’ holobiont natures [38,84,85,86]. However, preliminary observations of Bharadwaj et al. [38] indicated also that more robust plant genotypes can act more independently on microbiota assistance.

Our studies suggest that developing functional marker-assisted rice breeding or genetically engineered respectively edited rice plants by targeting AOX and glycolysis/fermentation-related genes could be among the promising strategies to confer salt stress and sustain rice productivity. However, such strategies require considering carefully the following general and specific state-of-the-art insights: (1) AOX genes have shown to be highly polymorphic in exon regions and even more pronounced in non-exon regions [86,87,88,89,90,91,92,93] In general, causative polymorphic sites within a gene were found to have low degree of conservation and phenotypic variation in a target trait can be linked to a diverse sequence polymorphisms [94]. (2) The relevance of AOX is due to its link to coordinating early plasticity provoked by continuously acting, ever-changing diverse environmental conditions, where salinity is only one among many stressors. This flexibility can be expected to rely on allelic polymorphisms [93] and flexible switching between polymorphic sites [95] depending on environmental and metabolic conditions; thus, diversity in AOX genes might be a desired trait per se; (3) as also shown in the present research, AOX is embedded in complex networking contexts [96] that evolved in unique, complex systems/organisms. This also includes the level of cells and their unique context in the plant body-shaped tissue and organ landscapes and concerns cell-free spaces (apoplasts). Thus, the species-specific role of target cells for defined agronomic traits needs to be considered [97], and (4) gene technology and gene editing have technical obstacles (reviewed, e.g, [98]) because they require in vitro culture as a first step. However, this means applying strong stress [86,91,99] and typically requires tissue and cell disruption from established networks. In contrast to functional marker-assisted selection at the seed and plant level, this bears at least the risk of undetected somaclonal variations through epigenetic and genetic side effects, which might change intrinsic, deeper phenotype characteristics of the original plant that can escape breeders’ awareness due to their focus on restricted agronomic or quality selection criteria.

In conclusion, our data support the relevant involvement of alternative pathways and glycolysis/fermentation in the more efficient stress response observed in a salt stress-tolerant rice genotype. This response is primarily associated with adaptive ROS balancing by AOX (via *AOX1a* expression), effective tuning of the antioxidant system and, secondarily, rapid energy production (via fermentation). Both contribute to sustaining and optimizing respiration. We cannot exclude the possibility that this intrinsic feature observed for stress-tolerance performance could have been modified by host–endophyte interactions, as we confirmed the holobiont nature of both genotypes.

## Figures and Tables

**Figure 1 plants-11-02145-f001:**
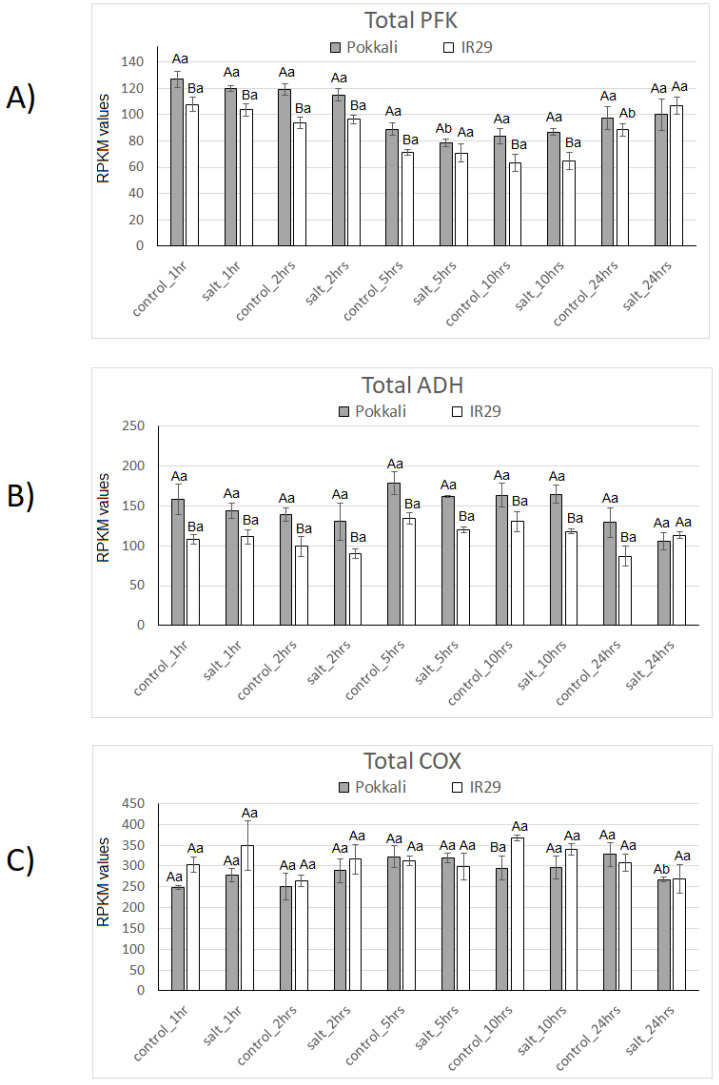
Expression profiles of Total *PFK* (**A**), *ADH* (**B**) and *COX* (**C**) genes in Pokkali and IR29 genotypes of *Oryza sativa* under salt stress. Data represent RPKM means with standard deviations from 3 biological replicates. For each gene, different capital letters indicate significant differences (at *p* < 0.05) between genotypes (Pokkali and IR29), while lowercase letters designate significant differences between treatments (control and stress) at the same time point and genotype, according to Bonferroni’s test.

**Figure 2 plants-11-02145-f002:**
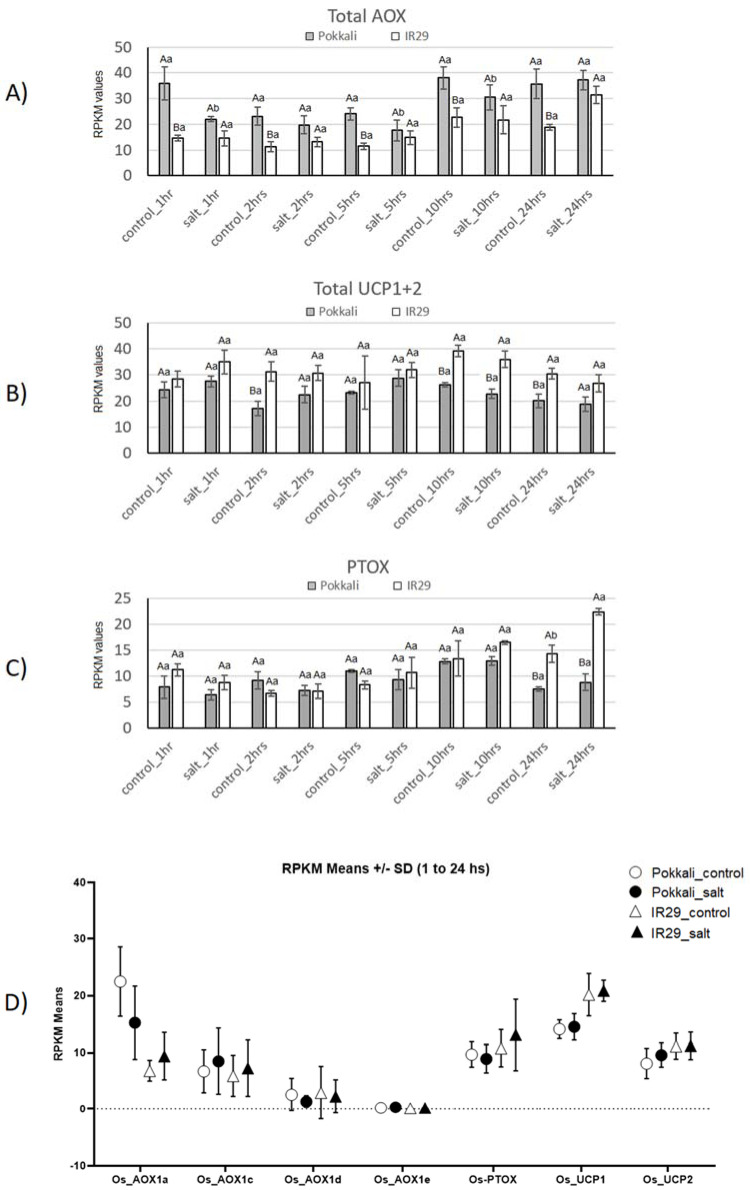
Expression profiles of Total *AOX* (**A**), *UCP* (**B**) and *PTOX* (**C**) genes in Pokkali and IR29 genotypes of *Oryza sativa* under salt stress. Data represent RPKM (**D**) means with standard deviations from 3 biolog-ical replicates. For each gene, different capital letters indicate significant differences (at *p* < 0.05) between genotypes (Pokkali and IR29), while lowercase letters designate significant differences between treatments (control and stress) at the same time point and genotype, according to Bonferroni’s test.

**Figure 3 plants-11-02145-f003:**
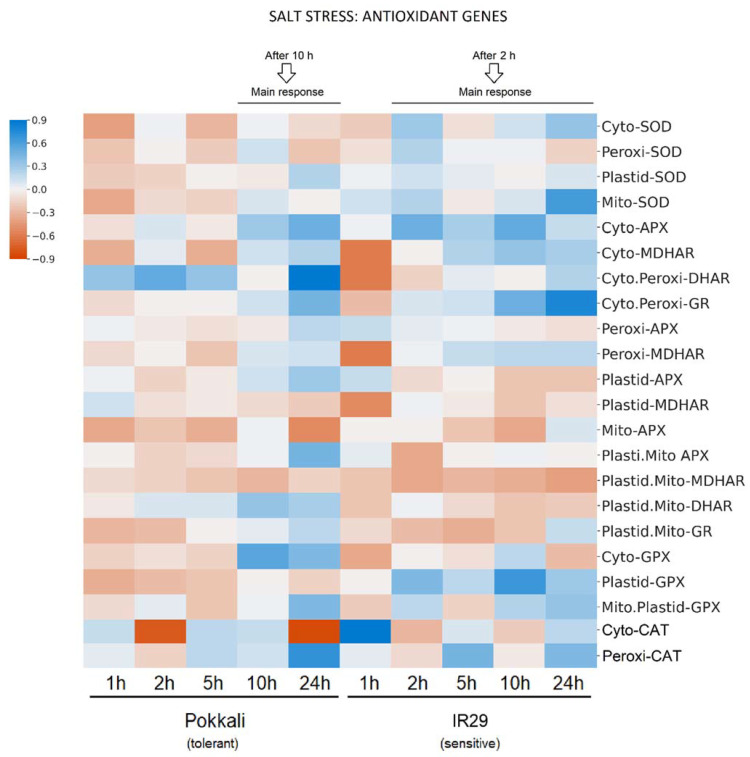
Heat map showing the gene expression of antioxidant enzymes in different cellular compartments of *Oryza sativa* genotypes under salt stress. The analyzed genes were APX, MDHAR, DHAR, GR, SOD, GPX and CAT in Pokkali and IR29 genotypes. The data represent log2 fold changes of salt treatment values at 1, 2, 5, 10 and 24 h in relation to the respective control conditions. In heat maps, the colors blue and orange represent up- and down-regulated genes, respectively. Statistical analyses of the RPKM means with standard deviations from 3 biological replicates are show in Appendix A.

**Figure 4 plants-11-02145-f004:**
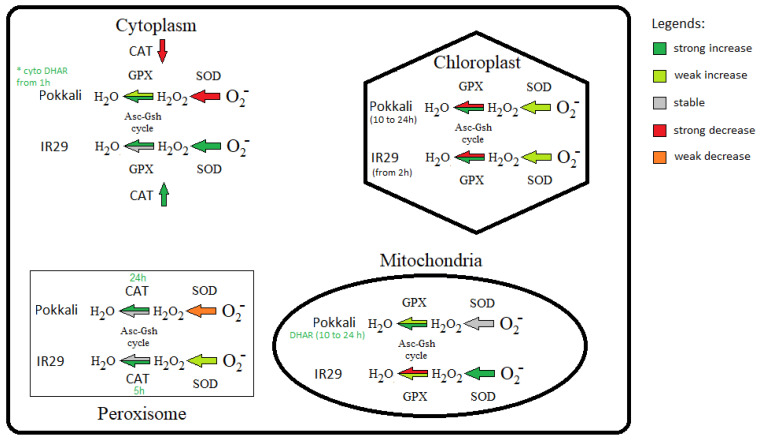
Schematic representation of antioxidant enzymes in different cellular compartments of *Oryza sativa* genotypes under salt stress. Significant values denoted substantial transcript increases or decreases, while non-significant values indicated weak increases or decreases according to Appendix A.

## Data Availability

The raw data used to obtain the results presented in this paper are available in SRA database from Genbank (NCBI) under bioproject numbers: PRJEB4671 (Pokkali) and PRJEB4672 (IR29).

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
