# Peer review of "Transcriptome Analyses in a Selected Gene Set Indicate Alternative Oxidase (AOX) and Early Enhanced Fermentation as Critical for Salinity Tolerance in Rice"

_plants, 2022, doi:10.3390/plants11162145_

Round 1
Reviewer 1 Report
The authors have made great efforts to make transcriptome analyses in gene expression related to ROS formation, impact ATP production and antioxidant system. The attained results are of interest for a better understanding of rice salinity tolerance. The authors can find some useful comments and suggestions to improve the quality of paper as below:
1. Reword the specific objectives of the study
2. Why did the authors select only 24h (1,2,5, 10 and 24h) for transcriptomic analysis?,
3. In this study, only two rice genotypes with contradictive traits (salt tolerance and none salt tolerance), how about the medium salt tolerance variety? and only two rice genotypes could reflect the reliable results?
4. Why were the rice materials stressed with only one salt concentration (300 mM NaCL)? hence it is difficult to conclude the gene expression output?
5. The current form of the manuscript was poorly organized and arbitrarily formatted and did not follow the style of the submitted journal, request to recheck and totally revise.
Author Response
Reviewer 1.
The authors have made great efforts to make transcriptome analyses in gene expression related to ROS formation, impact ATP production and antioxidant system. The attained results are of interest for a better understanding of rice salinity tolerance. The authors can find some useful comments and suggestions to improve the quality of paper as below:
Author`s comments
We would like to appreciate the thoughtful and stimulating comments and suggestions for improvements by the reviewer. We considered all comments and made a substantial effort to improve the manuscript. All main changes are marked in yellow all over the revised manuscript.
- Reword the specific objectives of the study
Author`s comments. Thank you for highlighting such a core subject, we changed the objectives in the abstract also in the end of introduction and hope is more conspicuous now.
- Why did the authors select only 24h (1, 2, 5, 10 and 24 h) for transcriptomic analysis?
Author`s comments. The selection purpose of earliest time points is to gain relevant knowledge on earliest responding genes/events linked to overall resilience on exposure to salinity. Our published results indicate predictability of plant resilience in the field from transcript profiles at early hours during cell reprogramming upon a diversity of stresses.
- In this study, only two rice genotypes with contradictive traits (salt tolerance and none salt tolerance), how about the medium salt tolerance variety? And only two rice genotypes could reflect the reliable results?
Author`s comments. We agree that at this stage our data support our hypothesis by two contrasting genotypes, but next an exhaustive validation is needed that should ideally include evaluation of genes from medium salt tolerant varieties .
- Why were the rice materials stressed with only one salt concentration (300 mM NaCl)? Hence, it is difficult to conclude the gene expression output.
Author`s comments.
We used public experimental data for our study. Our selection of available data focused on an adequate experiment with early time points for two contrasting genotypes with known performance in the field in search for support or rejection of our hypotheses. Further, as explained in our text (from line 291 to 294), severe salt stress is part of overall salt tolerance and this especially under higher temperatures and low relative humidity. So, we believe that we explained the context and carefully concluded that ‘our data suggest that AOX and ADH play a critical role during early cell reprogramming for improving salt stress tolerance by efficiently controlling ROS formation in mitochondria’ (abstract). However, we thank for making us aware and added now ‘can’ (can play).
- The current form of the manuscript was poorly organized and arbitrarily formatted and did not follow the style of the submitted journal, request to recheck and totally revise.
Author`s comments. A per the journal instructions to authors “Plants” now accepts free format submission.
Reviewer 2 Report
I read with great interest the article by Shahid Aziz et al entitled "Transcriptome analyses in a selected gene set indicate alternative oxidase (AOX) and 1 early enhanced fermentation as critical for salinity tolerance in rice". The authors evaluated a selected gene set (AOX, UCP, PTOX, ATP, PFK, ADH and COX) potentially involved in early cell reprogramming under stress in two rice genotypes contrasting in salinity response (Pokkali tolerant and IR29 susceptible). The approach enabled them to mark AOX and ADH as critical during early cell reprogramming and improve salt stress tolerance by efficiently controlling ROS formation in mitochondria. Thus, the authors claim AOX and ADH could be used as a candidate genes to engineer a salinity tolerant crops.
The concept is novel and interesting. English is good and well written. The article is worth publishing. However, there are several punctuation and writing errors. Therefore the manuscript needs to be thoroughly checked, and some minor suggestions should be considered before publication.
1. The introduction is too long. Try to lessen if possible.
2. Line 65-Replace 'ROS' by 'reactive oxygen species (ROS) as it appears for the first time.
3. Line 66 -The last sentence has repetitive information concerning the introduction. I
suggest other sentences containing a broad sense.
4. Line 186 - Information about the experimental confirmation of subcellular localization can be used in the results and discussion. But it is not necessary for the methodology.
5. Line 87 - replace 'reactive oxygen species (ROS)' by 'ROS'
6. Line 90 - what is means 'PQ'?
7. Line 130 - What is meant 'RNS'?
8. Given the wealth of transcriptomic data available in the database, why did you choose only 1 bioproject?
9. Is there any justification for selecting PFK and ADH genes specifically for glycolysis and fermentation pathways? Are key enzymes in the pathways?
10. Do you think 300 mM NaCl is sufficient to trigger salt stress in rice plants? Then, AOX is more powerful in ROS detoxification than the antioxidant system?
11. Line 227 - higher mRNA levels were detected in the ir29 . In which condition (control or treated)?
12. Line 252 - replace 'peroxisome' and 'decrease' by 'peroxisomal' and 'decreased.'
13. Line 255 - plastid-mito APX in chloroplast?
Author Response
Reviewer 2
I read with great interest the article by Shahid Aziz et al entitled "Transcriptome analyses in a selected gene set indicate alternative oxidase (AOX) and 1 early enhanced fermentation as critical for salinity tolerance in rice". The authors evaluated a selected gene set (AOX, UCP, PTOX, ATP, PFK, ADH and COX) potentially involved in early cell reprogramming under stress in two rice genotypes contrasting in salinity response (Pokkali tolerant and IR29 susceptible). The approach enabled them to mark AOX and ADH as critical during early cell reprogramming and improve salt stress tolerance by efficiently controlling ROS formation in mitochondria. Thus, the authors claim AOX and ADH could be used as a candidate genes to engineer a salinity tolerant crops.
The concept is novel and interesting. English is good and well written. The article is worth publishing. However, there are several punctuation and writing errors. Therefore the manuscript needs to be thoroughly checked, and some minor suggestions should be considered before publication.
Author`s Responses
We would like to appreciate the thoughtful and stimulating comments and suggestions for improvements by the reviewer. We considered all comments and made a substantial effort to improve the manuscript. All main changes are marked in yellow all over the revised manuscript.
- The introduction is too long. Try to lessen if possible.
Author`s comments. We agree that the section is a bit lengthy. Nevertheless, we have studied several pathways and feel that it is important to discuss each and in its complexity. Thus, we would like continuing with that current version.
- Line 65-Replace 'ROS' by reactive oxygen species (ROS) as it appears for the first time (Now, it is in the line 66).
Author`s comments. We made the change.
- Line 66 -The last sentence has repetitive information concerning the introduction. I suggest other sentences containing a broad sense.
Author`s comments. Thanks for pointing out the repetitive sentences, we removed the wordings.
- Line 186 - Information about the experimental confirmation of subcellular localization can be used in the results and discussion. But it is not necessary for the methodology.
Author`s comments. As per the reviewer`s suggestion we removed the word “Experimental” (See this in the line 194)
- Line 87 - Replace 'reactive oxygen species (ROS)' by 'ROS'
Author`s comments. We replaced “reactive oxygen species (ROS)' by 'ROS' (In this version, it is in line 88)
- Line 90 - What is means 'PQ'?
Author`s comments. PQ stand for Plastoquinone, we explained the abbreviation. (In this version, it is in line 90)
- Line 130 - What is meant 'RNS'?
Author`s comments. RNS stands for reactive nitrogen species, but as RNS is a common acronym, then we have used as such.
- Given the wealth of transcriptomic data available in the database, why did you choose only 1 bioproject?
Author`s comments. The Bioprojects (in fact, we used two bioprojects, one for each genotype) used are generated with same criteria for contrasting genotypes which is much helpful to explore the molecular level difference on exposure to salinity stress.
- Is there any justification for selecting PFK and ADH genes specifically for glycolysis stand it, but that the reader will understand it.and fermentation pathways? Are key enzymes in the pathways?
Author`s comments. Phosphofructokinase (PFK) is an enzyme highly regulated being its activity influenced by energy status in the cell. While the alcohol dehydrogenase (ADH) converts acetaldehyde to ethanol, acting in the terminal step of anaerobic glycolysis. Therefore, we believed that gene expression analysis of PFK and ADH could help in understanding about energy metabolism in rice under salt stress and how they could influence the mechanisms linked to adverse condition tolerance/adaption. In the manuscript, the information about ADH is given in line 179 “…ADH (Alcohol Dehydrogenase) genes [55] to denote total ADH in alcohol fermentation”. For PFK, we included the following sentence in lines 190 to 192 (highlighted in yellow).
- Do you think 300 mM NaCl is sufficient to trigger salt stress in rice plants?
Author`s comments. Rice is the most sensitive crop to salt stress. Supporting this information, previous studies showed that rice exposed to 60 mM NaCl developed symptoms of saline stress (Wairich et al., 2021) and significantly reduced grain yield to 50 mM NaCl (Yeo and Flowers, 1986). Therefore, 300mM NaCl is considered severe stress in lab experiments. Please see our explanations on salt stress from line 292 to 297.
Wairich, A.; Wember, L. S.; Gassama, L. J.; Wu, L. B.; Murugaiyan, V.; Ricachenevsky, F. K.; Pinheiro, M. M.; Frei, M. Salt resistance of interspecific crosses of domesticated and wild rice species. Journal of Plant Nutrition and Soil Science, v. 184, n. 4, p. 492-507, 2021.
Yeo, A. R.; Flowers, T. J. Salinity resistance in rice (Oryza sativa L.) and a pyramiding approach to breeding varieties for saline soils. Functional Plant Biology, v. 13, n. 1, p. 161-173, 1986.
- Then, AOX is more powerful in ROS detoxification than the antioxidant system?
Author`s comments. In fact AOX functions to avoid ROS formation, thus it acts before the detoxification event. In this sense, AOX would act as a first border defence to avoid ROS in the cell.
- Line 227 - Higher mRNA levels were detected in the ir29. In which condition (control or treated)?
Author`s comments. Higher UCP transcripts levels were observed in the ir29 genotype in both conditions: control and salt stress. Additional information is given in the text line 254 (new line position in this manuscript version)
- Replace 'peroxisome' and 'decrease' by 'peroxisomal' and 'decreased’ in the line 252 and plastid-mito APX in chloroplast in the line 256.
Author`s comments. We made the change. Now it corresponds to line 278.
Reviewer 3 Report
The report is well written, experimentation is well designed and the conclussions are sound and supported by the results. One of the main conclussions is that a more active glycolisis enhances stress tolerance. This was also observed in a recent report studying broccoli at the metabolomic level
https://bmcplantbiol.biomedcentral.com/articles/10.1186/s12870-021-03263-4
And I missed a comment of this fact in the discussion.
Minor points: ir29 or IR29? please use the same criteria throughout the manuscript.
Line 482, ref 17: please correct the format.
Author Response
Reviewer 3.
The report is well written, experimentation is well designed and the conclusion are sound and supported by the results. One of the main conclusions is that a more active glycolysis enhances stress tolerance. This was also observed in a recent report studying broccoli at the metabolomics level. https://bmcplantbiol.biomedcentral.com/articles/10.1186/s12870-021-03263-4
- And I missed a comment of this fact in the discussion.
Author`s comment. Thanks for this suggestion; however, the highlighted article does not focus on early cell reprogramming considering hours of stress (but six days) and its link to resilience prediction in the field, which is our approach. Furthermore, we believe that we sufficiently highlighted the role of glycolysis and the according representative gene all over the manuscript. Nevertheless, thanks for pointing to this interesting reference, we took notice.
Minor points: ir29 or IR29? Please use the same criteria throughout the manuscript.
Author`s comment. We are sorry for that, we changed all over the manuscript to ir29.
Line 482, ref 17: please correct the format.
Author`s comment. Thanks for highlighting, we revised the reference (In this manuscript version, this reference is in the line 500).
Round 2
Reviewer 1 Report
The authors have made great efforts to revise and edit following the comments and suggestions of reviewers. The paper is fine now, however, it still needs to recheck in grammar errors and format before publishing.
Reviewer 2 Report
Author has made all significant changes, therefore paper should be accepted in the current form.